# Nanoscale Bilirubin Analysis in Translational Research and Precision Medicine by the Recombinant Protein HUG

**DOI:** 10.3390/ijms242216289

**Published:** 2023-11-14

**Authors:** Paola Sist, Federica Tramer, Antonella Bandiera, Ranieri Urbani, Sara Redenšek Trampuž, Vita Dolžan, Sabina Passamonti

**Affiliations:** 1Department of Life Sciences, University of Trieste, Via Giorgieri 1, 34127 Trieste, Italy; psist@units.it (P.S.); ftramer@units.it (F.T.); abandiera@units.it (A.B.); 2Department of Chemical and Pharmaceutical Sciences, University of Trieste, Via Giorgieri 1, 34127 Trieste, Italy; rurbani@units.it; 3Institute of Biochemistry and Molecular Genetics, Faculty of Medicine, University of Ljubljana, Vrazov trg 2, 1000 Ljubljana, Slovenia; sara.redensek@mf.uni-lj.si (S.R.T.); vita.dolzan@mf.uni-lj.si (V.D.)

**Keywords:** bilirubin, fluorescent method, calibration, high-throughput assay, biomedical diagnostics, biomarker

## Abstract

Bilirubin is a toxicological biomarker for hemolysis and liver diseases. The current automated diazo method used in clinical chemistry has limited applicability in rodent models and cannot be used in small animals relevant to toxicology, microphysiological systems, cell cultures, and kinetic studies. Here, we present a versatile fluorometric method for nanoscale analysis of bilirubin based on its highly specific binding to the recombinant bifunctional protein HELP–UnaG (HUG). The assay is sensitive (LoQ = 1.1 nM), accurate (4.5% relative standard error), and remarkably robust, allowing analysis at pH 7.4–9.5, T = 25–37 °C, in various buffers, and in the presence of 0.4–4 mg × L^−1^ serum albumin or 30% DMSO. It allows repeated measurements of bilirubinemia in murine models and small animals, fostering the 3Rs principle. The assay determines bilirubin in human plasma with a relative standard error of 6.7% at values that correlate and agree with the standard diazo method. Furthermore, it detects differences in human bilirubinemia related to sex and *UGT1A1* polymorphisms, thus demonstrating its suitability for the uniform assessment of bilirubin at the nanoscale in translational and precision medicine.

## 1. Introduction

Bilirubin is a tetrapyrrolic pigment produced by cellular heme catabolism catalyzed by the sequential reactions of heme oxygenase (HO) and biliverdin reductase (BVR). In animals, heme catabolism is crucial because it provides detoxification of heme, a pro-oxidant molecule, and recycling of iron [1]. Moreover, the HO reaction products have long-range effects, as carbon monoxide (CO) sustains adaptation to hypoxia [2], and biliverdin forms a redox pair with bilirubin, acting as a radical scavenger [3]. Heme oxygenase 1 (HO–1) is encoded by a gene that is induced by a number of chemical and physical stressors, phytochemicals, oxidative stress agents, and chemical toxins [4], leading to cyto-protection and enhanced survival. Due to these effects, HO–1 is regarded as a therapeutic target in various diseases [5]. HO–1 activity is analyzed indirectly as a fold change in transcript and/or protein expression. However, analysis of bilirubin in cells, tissues, blood, and other fluids would allow more precise phenotypic characterization of the effects of HO–1 modulation, and this goal would be more easily attained by the availability of simple, sensitive, and “green” analytical methods that do not use hazardous chemicals and advanced equipment. 

Bilirubin is a dicarboxylic acid that, in principle, may be considered a highly water-soluble molecule. Nevertheless, six intramolecular hydrogen bonds between the propionic carboxyl groups and the pyrrole/lactam functions fold the molecule into the ridge–tile 3D configuration, which makes it sparingly soluble in water [6] and insoluble in apolar solvents as well [7]. The reported solubility in water spans from 7 nM at pH 7.4 [8] to 100 nM [9], and increases to 0.2 mM with increasing pH, due to ionization of the acidic groups of the molecule [10]. Hydrogen-bond-breaking solvents, like alkali and dimethyl sulfoxide, are most effective in solubilizing bilirubin [8]. Finally, bilirubin is a photosensitive and chemically labile molecule [11,12]. 

Regarding bilirubin analysis in animal blood, it should be noted that bilirubin binds to serum albumin [13,14] and forms a reversible complex, leaving only a very small amount (0.01%) of unbound bilirubin. Therefore, bilirubin analysis in serum or plasma depends on the extraction of bilirubin from albumin in a suitable solvent. The most widely used methods for conjugated and total bilirubin analysis in clinical practice are spectrophotometric methods based on the diazo reaction, enzymatic methods using bilirubin oxidase, and the vanadate oxidase method [15], but all of them have several specificity and sensitivity limits [16]. Other advanced methods based on HPLC, surface-enhanced Raman spectroscopy, and electrochemistry [17,18,19] place high demands on instrumentation and associated operational capabilities, and require sample preparation, while fluorescent nanosensors offer attractive alternatives, although they need further improvement in selectivity and sensitivity [20]. A promising method based on the bilirubin-specific fluorophore UnaG has been developed [21], enabling the direct determination of unconjugated bilirubin concentration in human newborn serum. 

In view of exploiting UnaG for monitoring bilirubin levels in a variety of pre-clinical, in vitro experimental models requiring cells or organs to grow on certain extracellular matrix-like substrates, we have developed a unique recombinant bifunctional protein named HUG, consisting of the fusion product of human elastin-like polypeptide (HELP) and UnaG. This bio-engineered product retains both its UnaG–bilirubin-dependent fluorescence emission and its HELP-specific thermo-responsive behavior [22,23]. Thus, it represents a case of an expanded potentiality of UnaG for the direct analysis of bilirubin at the nanoscale level. 

In this study, we present an exhaustive characterization of the performance of the HUG-based assay of bilirubin under experimental conditions applied across the full spectrum of translational medicine. In the first part of this study, we show, for the first time, how to implement independent fluorescence calibration under a variety of relevant conditions for in vitro studies, whereby users do not need to rely on some commercial calibrators, which may not be suitable in all analytical conditions. The second part is the complete characterization of the HUG assay performance, providing limits of quantification, accuracy, and applicability in a variety of buffered solutions used in pre-clinical experimental models for translational medicine. The third and last part of our study is on the performance of the HUG assay for measuring bilirubin in human serum, showing its capacity to detect subtle differences related to sex or genetic polymorphisms, standing out therefore as an excellent and affordable method for translational precision medicine.

## 2. Results

### 2.1. Preparation and Quality Controls of Standard Bilirubin Solutions 

#### 2.1.1. Principles

The principle of the procedure (Figure 1) is to prepare a minimum volume of 5 mM bilirubin (BR) dissolved in DMSO (stock or solution A), which is considered the best solvent [8], then dilute it to 10 µM in phosphate-buffered saline (PBS) (pre-calibrator or solution B1), and finally dilute the latter to nM solutions in PBS (solutions C1 or calibrators). Solutions B2 and C2 contain bovine serum albumin (BSA) to ensure stability of BR. 

The standard solutions were tested for both concentration and stability over time, by applying quality controls as summarized in Table 1, which specifies the reference solvents used. Other solvents were also tested for comparison purposes. 

#### 2.1.2. Validation of Bilirubin Concentrations in Standard Solutions

The concentration of the stock solution was verified by diluting it to 10 µM BR in PBS supplemented with 4 g × L^−^^1^ BSA. This solution was analyzed by the diazo reaction, according to a benchmark colorimetric method, using the recommended specific molar extinction coefficient (ε) of 76,490 cm^−1^ × M^−^^1^ [24]. The results confirmed that the estimates of bilirubin concentration in the pre-calibrator solutions (expected BR = 10 µM) or the diazo method were in agreement (BR = 10.3 ± 0.6 µM, n = 3).

To facilitate quality control of the pre-calibrator solution, quantitative determination of bilirubin was performed by direct UV–VIS spectroscopy. For this purpose, the UV–VIS analytical parameters of µM bilirubin solutions in DMSO, PBS, or PBS with either bovine serum albumin (BSA) or human serum albumin (HSA) were determined from UV–VIS spectra (Figure 2) and are listed in Table 2.

Both hyperchromic and bathochromic (redshift) effects were observed when albumin was added to the PBS solution. The molar extinction coefficient (ε) of bilirubin in PBS was lowest, but increased by about 40% when 4 g × L^−^^1^ BSA was added. Moreover, the spectroscopic LoD was 840 nM in PBS and 350 nM in PBS−BSA, showing a remarkable increase in sensitivity, probably due to the binding of bilirubin to albumin and an associated chemical stabilization effect. Concerning serum albumin, the difference in molar absorption coefficient of bilirubin standard solutions in BSA or HSA was negligible. Therefore, BSA was chosen for the preparation of all bilirubin standard solutions because of its affordable cost. By adding BSA to PBS, the molar extinction coefficient of BR was essentially the same as that obtained with DMSO and was considered to be the highest value achievable in water (Table 2).

Considering that the BR•BSA complex has a Kd ≈ 10^−^^6^–10^−^^7^ M [25,26], the estimated unbound bilirubin concentration is below the LoD of this measurement. Consequently, direct UV–Vis analysis of bilirubin in PBS–BSA is a measure of the BR•BSA complex [27] and not of total bilirubin (BR•BSA + BR). Nevertheless, analysis of a pre-calibrator solution by the standard diazo-based colorimetric method [24], which measures total bilirubin, perfectly correlated with the direct UV–VIS analysis (Figure 3) and demonstrated that BR•BSA can be considered the total bilirubin within the analytical limits of both direct UV–Vis spectroscopy and diazo-based colorimetry. 

To assess the effect of variable BSA concentrations below 40 g × L^−^^1^, we recorded the absorption spectra of 10 µM bilirubin in PBS containing 0.04 to 40 g × L^−^^1^ BSA, the latter value being the normal level of serum albumin in human blood [28]. Data showed that solutions containing 4 or 40 g × L^−^^1^ BSA had the same A_max_ = 465 nm and superimposable spectral shapes, whereas when the BSA concentration was below 4 g × L^−^^1^, the shape of the spectrum changed, the intensity of the peak decreased, and a blue shift (Amax = 445 nm) was observed (Figure 4).

#### 2.1.3. Stability of Bilirubin Standard Solutions

The stability of bilirubin in standard solutions is critical to the quality of the final nanoscale analysis. As part of the quality control process, the stability of each standard solution was validated, taking into account its concentration-specific usage properties.

Stability of the 5 mM stock solution

The 5 mM bilirubin solution in DMSO was aliquoted and stored at −20 °C. Aliquots were thawed and used to prepare 10 µM solutions in PBS (containing 0.2% DMSO), and their UV–Vis spectra were immediately analyzed. The spectra (Figure A1) confirmed the stability of bilirubin in DMSO for at least 4 months (A_440_ = 0.412 ± 0.007, n = 5); 

Stability of 10 µM pre-calibrator solutions

The stability of 10 µM BR solutions in PBS and PBS with 4 g × L^−^^1^ BSA was analyzed by UV–Vis for their specific A_max_ values at different time points up to 120 min as this is the time usually required for repeated bilirubin analyses in arrays of biological samples. Identical solutions in DMSO and 0.1 M NaOH were used as reference since these solvents ensure complete BR solubility [8,29]. The UV–Vis data (Figure 5a) showed that bilirubin was stable in DMSO for up to 2 h. When bilirubin was dissolved in PBS containing 4 g × L^−^^1^ BSA, the initial absorbance increased by about 10% in the first 30 min and reached the same constant value as in DMSO. When bilirubin was dissolved in PBS, A_440_ decreased by 21%, from 0.452 to 0.357, in 120 min. Dissolving bilirubin in alkali resulted in an initial higher A_max_ than in PBS, but thereafter the absorbance decreased. Similar results were obtained when the same tests were repeated with 1 µM BR (Figure 5b).

The delay to reach A_max_ was not observed when 1 µM BR was dissolved in PBS–BSA, indicating that the formation of the BR•BSA complex was complete at the time of analysis. In addition, A_max_ was higher than in DMSO, demonstrating the optimal function of albumin as a solvent additive. 

### 2.2. Nanoscale HUG-Based Fluorometric Assay of Bilirubin

#### 2.2.1. Basic Requirements

Concentration of HUG

The nanoscale analysis of bilirubin was performed using a fluorometric method based on high-affinity BR binding by the recombinant protein HUG that leads to a strong fluorescent signal [22]. This assay requires a non-limiting HUG concentration in the range of bilirubin analysis (0.05–50 nM). Experiments showed that 0.05 g × L^−^^1^ HUG was the concentration necessary and sufficient to obtain linearity in the desired range, and further increasing the polymer did not improve the signal (Figure A2);

Concentration of BSA

The influence of BSA concentration on fluorescence emission by the BR•HUG complex was examined in an experiment where BSA was added in a wide range of concentrations (0.4–40 g × L^−^^1^) in calibration tests in the range 0–50 nM bilirubin, and their angular coefficients were calculated. Data showed that this assay is independent of BSA concentration (Table 3). This result is very important because the same calibration curve obtained with calibrator solutions with 0.4 g × L^−^^1^ BSA can be used to determine the concentration of bilirubin in samples with variable and unknown BSA concentration.

#### 2.2.2. Linear Range of the Fluorometric Assay

The linearity of the assay was assessed employing serially diluted standard bilirubin solutions in the range 0–250 nM. They were prepared in PBS without or with 0.4 g × L^−^^1^ BSA, in order to verify any possible BSA interference in an extended bilirubin concentration range. The concentration of 0.4 g × L^−^^1^ BSA was selected since the detection of bilirubin in human serum by the HUG assay can be performed after diluting the samples 100- to 200-fold. 

The fluorometric intensity of bilirubin standard solutions in PBS was determined by adding HUG immediately after preparation of these solutions, obtaining a linear response up to about 50 nM (Figure 6). At higher concentrations, no increase in fluorescence was observed for standards prepared in the absence of BSA, whereas a substantial though nonlinear increase was seen in the presence of 0.4 g × L^−^^1^ BSA, presumably due to the BSA solubilizing effect. Under these conditions, the best linear regression parameters were obtained at bilirubin concentrations <50 nM and 75 nM, in the absence or in the presence of BSA, respectively.

#### 2.2.3. Optimal Reaction Time and Temperature

Figure 7 shows the progress of the fluorophore formation at 5 to 50 nM bilirubin. In PBS, the reaction was complete after 1 h, but it took 2 h if PBS contained 0.4 g × L^−^^1^ BSA. It is noteworthy that the steady-state fluorescence intensity was >30% higher in the presence of BSA, likely due to its property of preventing bilirubin loss from the solution.

The temperature effect on the HUG-dependent fluorescence emission in the range 0–50 nM bilirubin was assessed at 25 °C and 37 °C, as often used in biological and enzymatic assays. Data showed no temperature effect on fluorescence emission (angular coefficient = 756 ± 9 and 773 ± 21 at 25 °C and 37 °C, respectively, in the presence of 0.4 g × L^−^^1^ BSA).

#### 2.2.4. Limits of Nanoscale Bilirubin Detection and Quantitation

The assay sensitivity was characterized in the range 0–50 nM using standard bilirubin solutions containing 0.4 g × L^−^^1^ BSA. The fluorescence signal was linearly related to bilirubin concentration (Figure 8), and this result was consistent across 40 calibration curves. 

The application of the HUG assay to samples that may have widely varying bilirubin concentrations prompted us to evaluate the limits of detection in two different calibration ranges, with upper limits at 10 or 50 nM, respectively.

Bilirubin standard solutions in the range 0.05–50 nM were added to 10 µL HUG (1 g × L^−1^ in PBS) deposited in each well of the 96-well plate. Fluorescence was measured and data were fitted by linear regression analysis. The angular coefficient was used to calculate the sample concentration. The limit of detection (LoD) and limit of quantification (LoQ) were derived from the parameters of the standard curve, taking into account the standard error (SE) and the linear coefficient (slope, S) of the calibration curve according to the equations [30]:LoD = (3.3 × SE)/S
LoQ = (10 × SE)/S

The derived values of the limit of detection and limit of quantification for bilirubin are reported in Table 4.

#### 2.2.5. Solubility and Stability of Nanoscale Bilirubin Solutions

The need to collect and preserve biological samples prior to bilirubin determination prompted us to investigate in more detail the influence of putative stabilizing agents, such as BSA, HUG, HELP (human elastin-like polypeptide, i.e., the scaffold domain of HUG [22]), and the surfactant Triton X–100. Thus, freshly prepared standard pre-calibrator solutions (10 µM BR) in PBS alone or supplemented with stabilizers, such as 4 g × L^−^^1^ BSA or HUG or HELP or 1% Triton X–100, were used to obtain four nanoscale bilirubin calibrators (0, 5, 10, 25, and 50 nM). The fluorescence intensity of these calibrators was measured immediately or after storage at 4 °C for 24, 48, or 72 h (see Figure A3). 

To obtain a rigorous control of bilirubin stability in the explored nanomolar range, calibration curves were constructed at each time point studied, and their angular coefficients (nM^−^^1^) were calculated. Figure 9 shows the time-dependent change in these angular coefficients. The data showed that all three proteins tested optimally solubilized the pigment. However, only BSA and HUG preserved the pigment for 72 h, whereas HELP failed to prevent complete loss of signal at the end of the test. This confirms that the HELP domain of HUG does not specifically interact with bilirubin [22], but only has an initial positive solubilizing effect. A similar effect was observed with Triton X–100, as the initial fluorescence value was the same as for the protein-containing solutions. However, a partial loss of bilirubin was observed during storage. 

In the absence of these additives, bilirubin was not efficiently dissolved, as evidenced by the lower angular coefficient in PBS alone even at time zero, and substantial decay occurred within 24 h. When HUG was added only to the calibrator solutions and not to the 10 µM pre-calibrator standard, the soluble fraction of bilirubin remained constant, but an insoluble fraction could not return to solution and form a fluorescent HUG–bilirubin complex. 

#### 2.2.6. Accuracy and Precision

The accuracy of the method was evaluated according to guidelines to be followed for regulatory submissions [30,31,32]. We analyzed the bilirubin concentration of four different standard solutions (5–10–25–50 nM), and each standard solution was distributed in four wells (n = 4). Analysis was repeated three times, using fresh solutions (n = 3). This protocol was implemented on two different days. The relative error (%) between the measured values and the actual values of the calibrator standard solutions and their estimated concentration was evaluated. Day-to-day variation in HUG analysis was measured for all bilirubin solutions prepared with 0.4 g × L^−^^1^ BSA (Table A1), and accuracy assessed by relative error ranged from 1% to 9% with a median value of 4.5%.

The precision of the method was evaluated by analyzing four different bilirubin standard solutions (5–10–25–50 nM) in six replicates on two consecutive days. Precision was measured by the dispersion of individual results from the mean, expressed as the standard deviation or coefficient of variation (CV) of a series of measurements. The HUG assay coefficients of variation (CV) ranged from 1.7% to 5.8% (median = 2.6%) for BSA solutions (Table A2). The highest variability was observed at the lowest concentrations.

#### 2.2.7. Robustness

We measured the effects of incubation time, temperature, sample preparation, buffer pH, and other additives, thus simulating experimental conditions applied in pre-clinical in vitro models (molecular interactions; enzyme and receptor kinetics; cell and organ cultures). 

The fluorescence intensity of the BR•HUG complex in PBS was analyzed at pH 7.4, 8.5, and 9.5 (Table 5), finding that the pH had no influence on the signal, in this range, confirming previous observations with the BR•UnaG complex [33]. 

A number of other solvents for standard bilirubin solutions were tested for their influence on the fluorescence emission of bilirubin solutions in the range 0–50 nM (Table 5). Data showed that the assay can be performed in various buffered solutions in addition to PBS, such as Tris pH 8.0 and Hepes pH 7.4. By contrast, the Hanks’ Balanced Salt Solution resulted in fluorescence quenching. Supplementation of PBS with detergents, such as Triton X–100 and sodium taurocholate, but not Tween 20, was tolerated. 

DMSO could be added up to 30% (vol:vol; Figure A4). The solutions of bilirubin in 0.4 g × L^−^^1^ BSA or HSA and PBS–Triton 1% (pH 7.4) achieved the highest sensitivity, i.e., the ability to distinguish small differences in the concentration of the analyte. Remarkably, BSA can replace HSA, which results in reducing the operating costs of this assay. Thus, this assay can be applied to different experimental conditions in biomedical research. 

### 2.3. Direct Analysis of Bilirubin in Human Plasma by the HUG Assay

We tested the performance of this assay in measuring differences in the bilirubin levels in human plasma samples. We wanted to challenge this assay for its capacity to provide a uniform determination of unconjugated bilirubin across the entire span of translation medicine, from in vitro models to precision medicine.

First, we performed a spike-and-recovery experiment to determine whether there was a matrix effect, finding no significant difference between the expected and observed values (Table A3). 

Then, we evaluated the assay precision on the plasma bilirubin concentration as the relative standard deviation, calculated for 15 different samples analyzed on different days (46 total measurements). The mean % RSD was 6.7 (min = 1.9 and max = 12.6), a value ≤15%, as recommended in analytical guidelines [34,35]. 

Finally, we evaluated the correlation between plasma bilirubin concentrations determined by the standard diazo reaction-based protocol [24], which measures total bilirubin (sum of bilirubin and its bilirubin glucuronide) and the HUG assay, run in the presence of β–glucuronidase. The 34 samples were randomly taken from a set of 226 plasma specimens (bilirubin concentration range 1.5–22.6 µM). Data in Figure 10a show that correlation was 0.84. Considering the precision of the HUG assay and the full recovery of spike nM bilirubin in plasma, we conclude that the diazo test overestimates total bilirubin, a well-known fact [24]. The Bland–Altman scatter plot [24] (Figure 10b and Table A4, with an exhaustive presentation of the features of the Bland–Alman analysis) shows that the measurements in the bilirubin physiological range had a bias of −0.99 µM units, reflecting the fact that the diazo assay overestimates bilirubinemia due to interfering compounds in human serum [28] consistently with the <1 correlation coefficient (Figure 10a).

The range between the upper and lower LoA values was 3.69 µM, which can be expected from comparing two methods with very different LoQ parameters and, therefore, operating conditions (as specified in Materials and Methods). Nevertheless, all data (except for one) were within the upper and lower limits of agreement. Altogether, the two methods agree, but small differences in intra- or inter-individual bilirubinemia should be assessed by one method only. This is most important, since an inverse relationship between increasing bilirubinemia and cardiovascular disease has been described within the normal physiologic range of bilirubin levels (median = 10 µM) [36].

Indeed, the power of the HUG assay to discriminate well-known physiological, group-related differences in bilirubinemia was tested by analyzing plasma collected from 224 subjects. In a group of 224 included subjects, there were 127 (56.7%) males and 97 (43.3%) females, with a median age of 62.0 years (54.8 years–71.7 years). Genotypes of all six studied SNPs were in Hardy–Weinberg equilibrium (Table A5). Data showed that the HUG assay identified differences of median plasma bilirubin of 13%, if related to sex [37], and 22% or more, in the case of polymorphisms at the level of the *UGT1A1* gene [38] (Table 6). Females presented with lower bilirubin plasma concentrations. On the other hand, higher plasma bilirubin concentrations were observed in carriers of an additional *UGT1A1* rs8175347 TA repeat. Both associations were already reported in the literature, which adds a new level of validity and credibility to our method. We observed no associations with any other tested SNPs (Table A5).

Considering that the error of the HUG assay is 6.7%, in can be calculated that the sample size needed in each group to detect sex-related differences in bilirubinemia, attaining statistical significance of *p* < 0.01 with a 95% probability, is 10.

## 3. Discussion

### 3.1. Advantages and Limitations of the HUG Assay

HUG is a recombinant fusion protein, formed by two functional domains. Its UnaG domain retains both the bilirubin-specific fluorophore activity already demonstrated for UnaG [33] and the thermo-responsive properties of HELP [39]. With respect to UnaG (Kd = 10^−^^10^ M), HUG has a lower affinity for bilirubin (Kd = 10^−^^9^ M), but this is not a limitation because it is still higher than BSA or HSA [23], resulting in full extraction of unconjugated bilirubin from the albumin pool. The albumin independence of the assay allows flexibility in diluting human serum/plasma samples, and extends the applicability domain of this assay to sera or plasma of other animals, for veterinary and biomedical research.

The HELP domain of HUG acts by protecting UnaG from unfavorable solvent effects, as demonstrated by its fluorescence emission stability in the presence of a variety of additives and, in particular, the aprotic DMSO co-solvent up to 30%. By contrast, UnaG tolerates no more than 0.2% DMSO [40]. The HELP domain provides a number of other technological potentialities, yet to be fully exploited. One of the major ones is the possibility to create a 3D matrix by inducing polymer cross-linking [22], serving as a cell culture substrate.

Unlike other assays so far described in the literature [18] and the UnaG assay itself [21], this study on HUG as a bilirubin-specific bio-receptor provides an accurate calibration procedure which supports reproducible results or effective troubleshooting. The HUG assay makes it feasible and affordable to perform studies of bilirubin interaction with molecular targets in the most basic enzyme or receptor kinetics settings. Its LoQ as low as 1.1 nM and its excellent parameters of accuracy and precision (4.5% and 2.6%, respectively) ensure high-quality data. Other applications are investigations of heme metabolism in cell cultures [41] or transport of bilirubin in isolated and perfused rat liver [42], as examples of pre-clinical models in translational research. 

A distinct advantage of bilirubin assays exploiting UnaG is that there is no interference by ditauro bilirubin, biliverdin, urobilin, hemoglobin, and lipids, as demonstrated by pioneering works with UnaG [33]. By using HUG, we have confirmed these findings and expanded the spectrum of non-interfering molecules, such as estradiol 17–beta glucuronide and its aglycone, pravastatin, taurocholate, cyanidin 3–glucoside, malvidin 3–glucoside, peonidin 3–glucoside, and resveratrol [42].

The main limitation of this assay may be the availability of the HUG recombinant protein, which is not commercially available. Though the procedure for its lab-scale production is basic and open, this may limit the widespread use of this assay. Predictably, this limitation will be overcome as soon as high-quality research results are published.

### 3.2. Domains of Application in Precision Medicine

Small changes of bilirubinemia are related to a significant decrease in mortality and disease risk [43], but only a few factors are known for causing mild upwards modulation of bilirubinemia, e.g., fasting [44] or weight loss [45], and the interest in finding drugs or natural compounds capable of inducing “iatrogenic Gilbert’s syndrome” [46] is still elevated [47]. Precision medicine requires high-performance analytical methods. The precision of this assay is given by its mean relative standard error of 6.7% (min 1.9%–max 12.6%) in human plasma analysis. It can be calculated that the number of subjects needed to detect 10% differences in mean bilirubinemia between groups is 15; by considering the precision worst-case scenario (13% relative standard error), the sample size increases to 62. This assay can therefore serve the task of assessing the impact on bilirubinemia of drug metabolism or other modifiers (e.g., diets, feeding schemes, nutraceuticals, stress, sports, and many others) in small pilot studies whose outcomes are necessary to design larger clinical trials or population studies. Similarly, it is a practical tool to perform targeted screening of biobanks of completed clinical trials where bilirubinemia was not planned to be measured. This assay is an excellent companion diagnostic method for the development of a deeper understanding and exploitation of human biomarkers.

## 4. Materials and Methods

### 4.1. Materials

Analytical-grade chemicals purchased from Merck Life Science S.r.l. (Milano, Italy) were: bilirubin (BR, purity 99%, lot. 031M1429V #B4126), bovine serum albumin fraction V (BSA, A–7906, purity >98%), human serum albumin (HSA, purity 97–99%, A9511), β–glucuronidase (50180211), dimethyl sulfoxide (DMSO), dibasic sodium phosphate (Na_2_HPO_4_), monobasic sodium phosphate (NaH_2_PO_4_ × H_2_O), sodium chloride (NaCl), sodium hydroxide (NaOH), hydrochloric acid (HCl), sulphanilic acid (NH_2_C_6_H_4_SO_3_H), caffeine, sodium nitrite (NaNO_2_), sodium acetate (CH_3_COONa), sodium benzoate (C_6_H_5_COONa), EDTA disodium salt (C_10_H_14_N_2_Na_2_O_8_ × 2H_2_O), potassium sodium tartrate (KNaC_4_H_4_O_6_), Triton X–100 (T8787), Na–Taurocholate (86339), Hanks’ Balanced Salt Solution (HBSS, 55037C), Tween 20 (P1379), Trizma^®^ Base (T–6066), HEPES (H3375). Ultrapure water milliQ was used to prepare each solution. HUG was synthesized and purified as described [22]. Black 96-well microplates were used (Nunc^®^, purchased by Fisher Scientific Italia, as part of Thermofisher, Segrate, Milano, Italy, code 237107; polystyrene, sterile, non-treated surface).

### 4.2. Standard Bilirubin Solutions 

The principle of the procedure (Figure 1) is to prepare a minimum volume of 5 mM bilirubin (BR) dissolved in DMSO (stock or solution A), which is considered the best solvent [8], then dilute it to 10 µM in phosphate-buffered saline (PBS) (pre-calibrator or solution B1), and finally dilute the latter to nM solutions in PBS (solutions C1 or calibrators). Solutions B2 and C2 contain bovine serum albumin (BSA) to ensure stability of BR. All bilirubin solutions were prepared under dim light in a dark room and stored in brown bottles until analysis. 

#### 4.2.1. Stock Solution

The bilirubin (BR) stock solutions were prepared by weighting the dry powder and dissolving it in DMSO to a concentration of 5 mM. These solutions were stored at −20 °C.

#### 4.2.2. Pre-Calibrator Solutions

Pre-calibrator solutions were prepared by diluting the stock solution to 10 µM BR in PBS (phosphate-buffered saline, PBS, pH = 7.4) without or with BSA. When specified, 1 µM BR solution was obtained by diluting the 10 µM BR solution in PBS without or with BSA.

#### 4.2.3. Calibrator Solutions 

Serial bilirubin solutions in the range of 0.05–50 nM were prepared by diluting the pre-calibrator solution. They were prepared under dim light in a dark room and stored in brown bottles until analysis. Further details were described in a technical protocol [48].

### 4.3. Spectrophotometric Measurements of Bilirubin Standard Solutions 

Absorption spectra of bilirubin solutions prepared by diluting 5 mM BR in DMSO (stock solution) to 10^−^^5^–10^−^^6^ M in various solvents (DMSO, PBS, 4 g × L^−^^1^ HSA or BSA) were recorded at λ = 350–600 nm in a double-beam spectrophotometer (CARY–4E UV–visible spectrophotometer, Cary Instruments, Monrovia, Calif. 91016) at T = 25 °C using quartz cuvettes (Suprasil 10 ± 0.01 mm, Helma Cells Inc., Jamaico, NY 11424, USA). Spectra were recorded immediately after preparation of the solutions. For solutions containing serum albumin, recording began after 30 min. All measurements were performed in triplicate. 

### 4.4. Colorimetric Measurements of Bilirubin Standard Solutions

Bilirubin solutions in PBS containing 4 g × L^−^^1^ BSA in the micromolar range were analyzed according to the protocol described by Klauke [24]. The diazo reagent was prepared by mixing an aqueous solution of sodium nitrite with a sulfanilic acid solution in a 1:50 ratio. 

Bilirubin standard solutions (0.2 mL; 1–100 µM) were mixed with the caffeine reagent (1.0 mL), the diazo reagent (0.5 mL), and the alkaline tartrate solution (1.0 mL). These components were added to the cuvettes (1 cm path length) in the above order, with a 10 min wait between each addition. After 5 min of incubation at 25 °C in the dark, the absorbance at λ = 598 nm was read in a double-beam spectrophotometer (CARY–4E UV–visible spectrophotometer, Cary Instruments, Monrovia, CA 91016, USA). The blank for each sample was obtained by replacing the diazo reagent with an equal volume of sulfanilic acid solution.

### 4.5. Fluorometric Measurements of Bilirubin Standard Solutions

The assay was carried out in black 96-well plates (Nunc^®^) loaded with 10 µL HUG (1 g × L^−^^1^ in PBS). Fixed volumes (0.2 mL) of serially diluted standard bilirubin solutions (0.05–50 nM) were added to the wells in 4 replicates. The microplate was incubated at T = 25 °C for 1 h (or for 2 h if in the presence of serum albumin) prior to fluorescence intensity measurement (λ_ex_ = 485 nm, λ_em_ = 528 nm; gain 100, reading height 2.50 mm; T = 25 °C) in a benchtop multiplate reader (Synergy H1; BioTek, Winooski, VT, USA). The mean fluorescence reading value of empty plate was about 20–30 Arbitrary Units (A.U.), while the intrinsic fluorescence of the solvent (0.4 g × L^−^^1^ BSA in PBS) was <100 A.U. and was always subtracted from the HUG–BR signal (1000–40,000 A.U.). All experiments were performed with freshly prepared solutions at room temperature.

### 4.6. Study Subjects and Ethics Statement

A total of 224 consecutive Parkinson’s Disease patients were included in the assessment of plasma bilirubin concentrations and in the analysis of single-nucleotide polymorphisms (SNPs) in selected genes from the bilirubin metabolic pathway. Patients were recruited at the Department of Neurology, University Medical Center Ljubljana, Slovenia, between October 2016 and April 2018 [49]. The study protocol was approved by the Slovenian Ethics Committee for Research in Medicine (KME 42/05/16 and 0120–268/2016/16). All subjects gave written informed consent in accordance with the Declaration of Helsinki. Peripheral blood, collected in K2 EDTA tubes, was processed within four hours after withdrawal. Blood was centrifuged at 2200× *g*, for 10 min at 4 °C, to separate plasma from the blood cells. Plasma was stored at −80 °C until bilirubin was measured, while blood cells were stored at −20 °C until DNA extraction.

### 4.7. Bilirubin Fluorometric Measurements in Human Plasma Samples

Human plasma (10 µL) was added to 1.990 mL HUG solution (0.05 mg × mL^−1^, pH 7.4) and then divided into two 1 mL aliquots, one for BR and the other for total BR (BR + BR glucuronide) quantification. For analysis of BR, volumes of 200 µL were added directly to the multiwell plate in four replicates. For analysis of total BR, 2.5 µL of β–glucuronidase was added to the second 1 mL aliquot (final concentration 0.0875 U × µL^−1^) and then distributed to the multiwell plate in four replicates. The microtiter plate was incubated at T = 25 °C, and fluorescence intensity was measured after 16 h. Intrinsic sample fluorescence was measured in diluted human plasma (5 µL in 1 mL PBS, pH 7.4), and the value (<200 A.U.) was subtracted from the HUG–BR fluorescence signal (1000–40,000 A.U.).

### 4.8. DNA Extraction and Genotyping

Genomic DNA was isolated using the FlexiGene DNA Kit (Qiagen, Hilden, Germany) in the course of our previous study [49]. Six SNPs (*BLVRA* rs699512, *UGT1A1* rs8175347, *HMOX1* rs2071746 and rs2071747, *HMOX2* rs2270363 and rs1051308) were genotyped with KASPar assays (KBiosciences, Unit 7 Maple Park Essex road Hoddesdon, Herts, UK, and LGC Genomics, Queens Road, Teddington, Middlesex, TW11 0LY, UK) according to the manufacturer’s instructions. In total, 10% of samples were genotyped in duplicate as quality control, and all the results were concordant.

### 4.9. Statistical Analyses 

All data were analyzed and plotted using GraphPad Prism 10.1.0 (264) (GraphPad Softwares, Boston, MA, USA). Unpaired Student’s *t*-test was performed using standard significance level α = 0.05, Estimation of LoD and LoQ values was performed using Excel Analysis ToolPak. 

Within the association analysis, median and 25th to 75th percentile range were used to describe central tendency and variability of continuous variables, while frequencies were used to describe the distribution of categorical variables. The agreement of genotype frequencies with Hardy–Weinberg equilibrium was assessed with chi-squared test. Nonparametric Mann–Whitney U test and Kruskal–Wallis test were used to assess the effects of sex and genotypes on plasma bilirubin concentrations. All statistical tests were two sided. *p*-values below 0.050 were considered statistically significant. All statistical analyses were carried out by IBM SPSS Statistics, version 21.0 (IBM Corporation, Armonk, NY, USA).

## 5. Conclusions

The HUG assay can be used for the uniform analysis of bilirubin in several distinct experimental models employed in the arc of translational and precision medicine. From the technological point of view, this study covers the progression across the Technology Readiness Level scale, from TRL 4 to TRL 6.

## Figures and Tables

**Figure 1 ijms-24-16289-f001:**
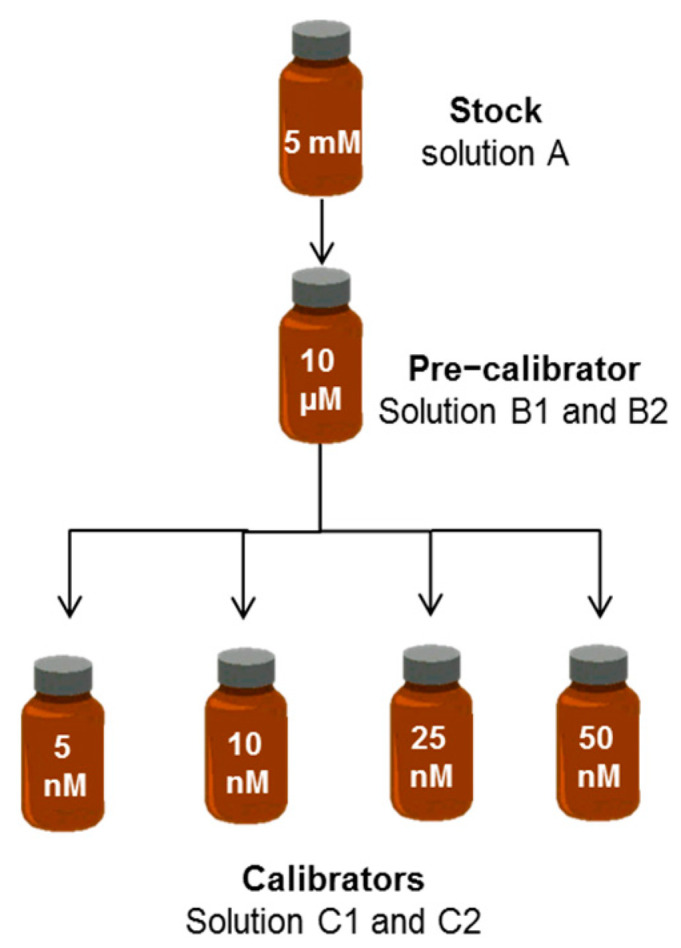
Preparation of standard bilirubin solutions. The solvent of solution A is DMSO. The solvent of solutions B and C is PBS without (B1 and C1) or with BSA (B2 and C2).

**Figure 2 ijms-24-16289-f002:**
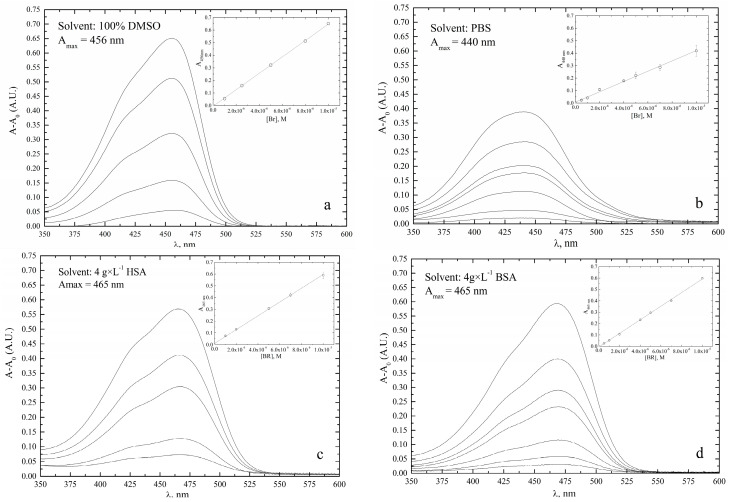
UV–visible spectra of 1−10 µM bilirubin in different solvents. (**a**) dimethylsulphoxyde (DMSO); (**b**) phosphate buffer (PBS), pH 7.4; (**c**) PBS with 4 g × L^−1^ human serum albumin (HSA), pH 7.4; (**d**) PBS with 4 g × L^−1^ bovine serum albumin (BSA), pH 7.4. Spectra were recorded at T = 25 °C using Suprasil quartz cuvettes 10 ± 0.01 mm in size.

**Figure 3 ijms-24-16289-f003:**
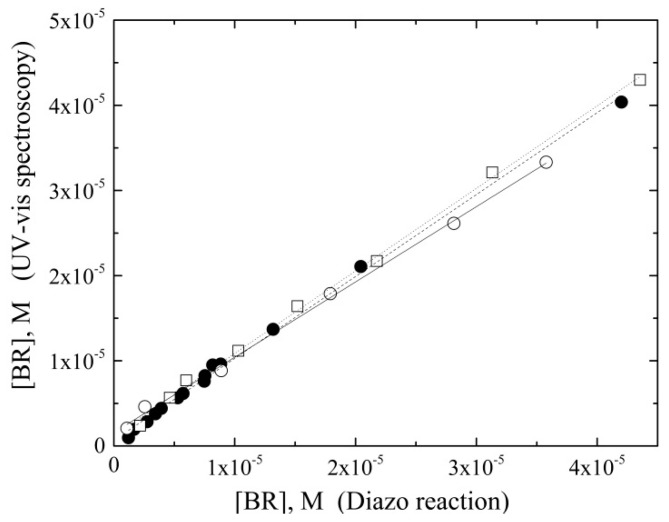
Correlation of bilirubin analysis by direct UV–Vis spectroscopy versus the diazo colorimetric method. Bilirubin was dissolved in three solvents and analyzed by both methods. Data were fitted to the equation y = mx + q: PBS (empty circles and continuous line; m= 0.884, q = 1.58 × 10^−^^6^, R^2^ = 0.998); PBS with 4 g × L^−^^1^ BSA (filled circles and dashed line; m = 0.962, q = 0.667 × 10^−^^6^, R^2^ = 0.997); 100% DMSO (empty squares and dotted line; m = 0.969, q = 1.18 × 10^−^^6^, R^2^ = 0.998).

**Figure 4 ijms-24-16289-f004:**
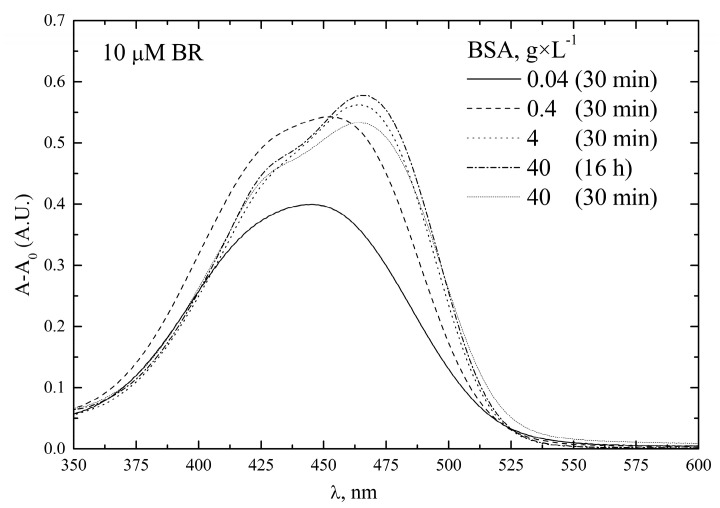
Effect of BSA concentration on the UV–Vis spectra of bilirubin. Bilirubin (BR) pre-calibrator solutions (10 µM) were prepared in PBS supplemented with different BSA concentrations. UV–Vis spectra were recorded at T = 25 °C after 30 min.

**Figure 5 ijms-24-16289-f005:**
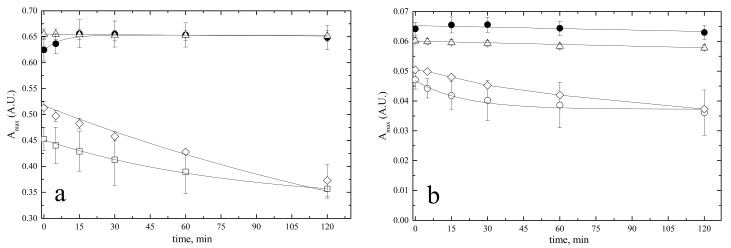
Stability of bilirubin pre-calibrator solutions. BR was dissolved in 100% DMSO (open triangles) or 0.1 M NaOH (open diamonds) or PBS (open circles) or PBS with 4 g × L^−^^1^ BSA (filled circles). Solutions were maintained at constant T = 25 °C for up to 2 h and analyzed by UV–Vis spectroscopy at *λ*_max_, indicated in Table 2. Solutions in NaOH were analyzed at *λ*_max_ = 425 nm. (**a**) 10 µM BR, (**b**) 1 µM BR.

**Figure 6 ijms-24-16289-f006:**
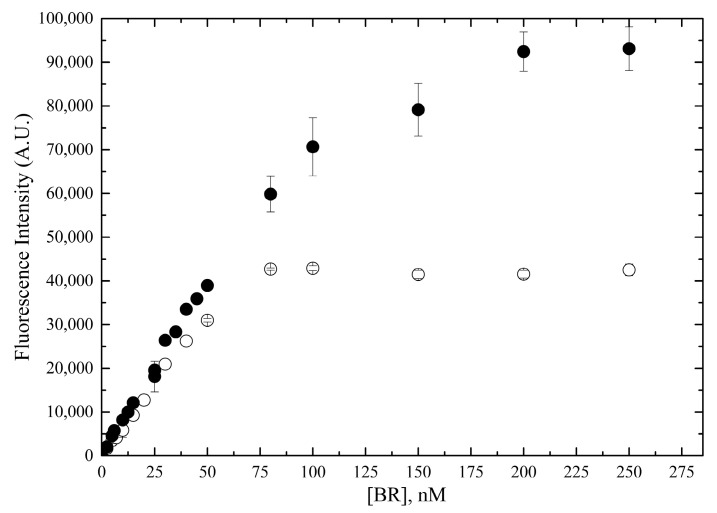
Linearity range of bilirubin-dependent fluorescence emission of HUG. Fluorescence intensity of BR•HUG complex in a wide range of bilirubin concentrations in PBS (open circles) or PBS with 0.4 g × L^−^^1^ BSA (filled circles). Experimental conditions: [HUG] = 0.05 g × L^−^^1^ (0.83 × 10^−^^6^ M), T = 25 °C, reaction time 1 h (PBS) or 2 h (PBS–BSA).

**Figure 7 ijms-24-16289-f007:**
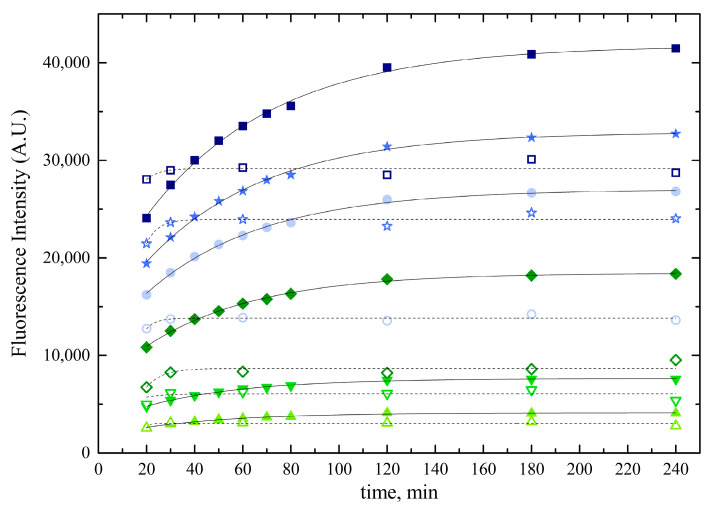
Progress of the bilirubin–HUG complex formation in the absence and in the presence of BSA. Bilirubin solutions at concentrations of 5 nM (up triangle), 10 nM (down triangle), 20 nM (diamond), 30 nM (circle), 40 nM (star), and 50 nM (square) were prepared in either PBS (open symbols and dotted lines) or PBS with 0.4 g × L^−^^1^ BSA (filled symbols and continuous lines). They were incubated with 0.05 g × L^−^^1^ (0.83 × 10^−^^6^ M) HUG at T = 25 °C. Fluorescence intensity of the BR•HUG complex was monitored for up to 4 h. Data were fitted to the single exponential equation.

**Figure 8 ijms-24-16289-f008:**
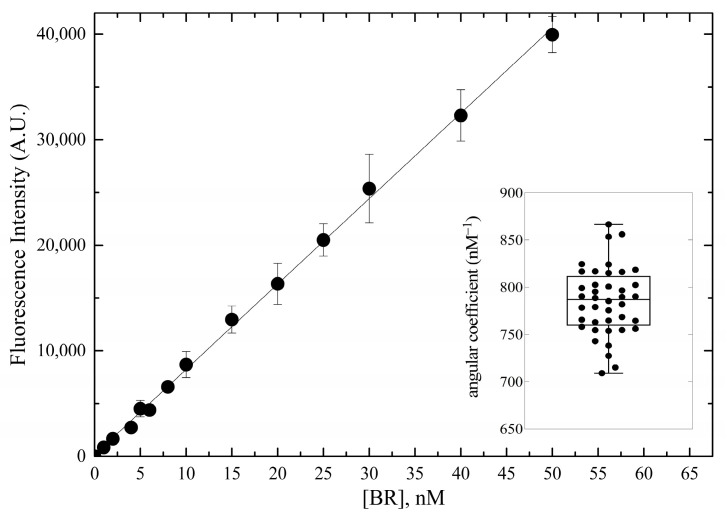
Calibration of bilirubin-dependent fluorescence emission of HUG. Nanoscale bilirubin solutions in PBS with 0.4 g × L^−^^1^ BSA were incubated with 0.05 g × L^−^^1^ (0.83 × 10^−^^6^ M) HUG at T = 25 °C for 2 h. Fluorescence intensity data (means ± SD, n = 40 for each tested concentration) were fitted to the equation y = mx + q (parameters: m = 785, q = 146, R^2^ = 0.999). The inset shows the box-plot analysis of the angular coefficients (*t*-test unpaired).

**Figure 9 ijms-24-16289-f009:**
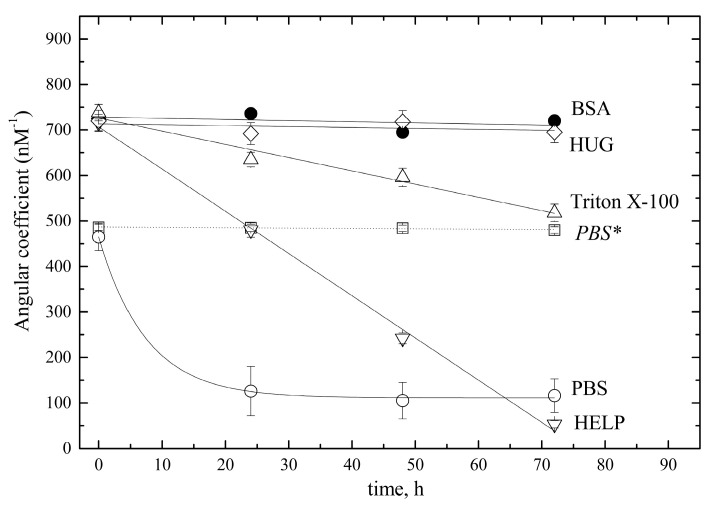
Stability of nanomolar bilirubin solutions in different solvents. Bilirubin solutions (5, 10, 25, 50 nM) were prepared in PBS (open circles) or in PBS supplemented with 0.4 g × L^−^^1^ BSA (filled circles) or 0.4 g × L^−^^1^ HUG (diamonds) or 1% Triton X–100 (triangles up), or 0.4 g × L^−^^1^ HELP (triangles down) and kept at T = 25 °C. Calibration curves and their respective angular coefficients (nM^−^^1^) were obtained at different times, up to 70 h. * The dotted line across open squares refers to standard solutions prepared in PBS and immediately supplemented with 0.05 g × L^−^^1^ HUG.

**Figure 10 ijms-24-16289-f010:**
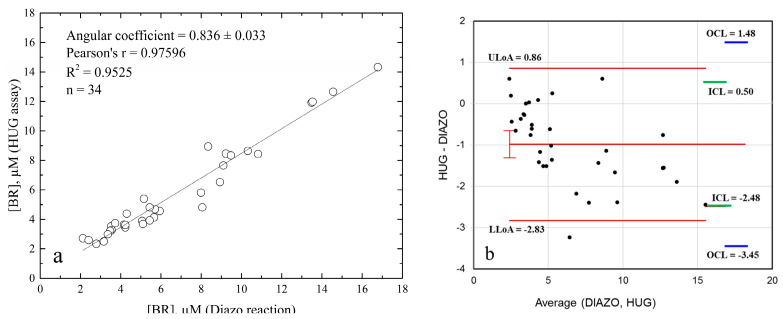
Comparison of total bilirubin analysis in human plasma samples by two assays. The analysis by the HUG assay was performed in the presence of β–glucuronidase (0.085 IU × µL^–1^). The analysis by the diazo reaction was performed in the presence of caffeine. (**a**) Correlation plot. (**b**) Bland–Altman plot, obtained by the software available in Carkeet, 2020. Red lines: Average of the two paired measurements (Diazo–HUG) and upper LoA (ULoA) and lower LoA (LLoA). Green lines: inner confidence limits (ICL). Blue lines outer confidence limits (OCL).

**Table 1 ijms-24-16289-t001:** Standard solutions of bilirubin (BR) and analytical approach to quality controls.

ID Term	[BR]	Solvent	Quality Control
A. Stock	5 mM	DMSO	UV–VIS
B. Pre-calibrator	10 µM	PBS–BSA	UV–VIS; diazo reaction
C. Calibrator	50 nM	PBS–BSA	HUG fluorescence

**Table 2 ijms-24-16289-t002:** UV–Vis spectroscopic results and derived parameters of bilirubin solutions in four solvents. Serially diluted solutions (0.5–10 µM) were prepared in 100% DMSO (*v*/*v*), PBS in the absence or presence of 4 g × L^−^^1^ BSA, or HSA. Solutions were freshly prepared from the standard bilirubin stock (5 mM in DMSO) and immediately analyzed by UV–VIS spectrometry. A_max_ was recorded in a 10 µM BR solution. All experiments were performed in triplicate.

	DMSO	PBS−BSA	PBS−HSA	PBS
Amax (A.U.)	0.631 (±0.02)	0.636 (±0.05)	0.601 (±0.02)	0.446 (±0.05)
λmax (nm)	456	465	465	440
ε (cm^−1^ M^−1^)	63,175 (±2859)	61,079 (±1940)	60,053 (±1425)	41,885 (±2558)
R^2^	0.9997	0.9992	0.9989	0.9946
LoD (nM)	240	350	426	840
LoQ (nM)	720	1100	1290	2600

**Table 3 ijms-24-16289-t003:** Fluorescence intensity of HUG–BR solution as a function of BSA concentration. Experimental conditions: T = 25 °C and t = 2 h.

[BSA] g × L^−1^ Pre-Calibrator	[BSA] g × L^−1^ Calibrators	Angular Coefficient (nM^−1^)
40	40	744 ± 74
4	4	815 ± 30
4	0.4	785 ± 36

**Table 4 ijms-24-16289-t004:** Limits of detection and quantitation of bilirubin in PBS with 0.4 g × L^−^^1^ BSA solutions (n = 6).

BR (nM)	LoD (nM)	LoQ (nM)
0.05–10	0.36	1.10
0.5–50	1.56	4.75

**Table 5 ijms-24-16289-t005:** Calibration parameters obtained with nanoscale standard solutions of bilirubin in various solvents. The reaction time was 1 h, except for the solutions containing BSA or HSA. T = 25 °C.

Solvent	Angular Coefficient (nM^−1^)	R^2^	Fluorescence Intensity of 50 nM BR Solution (A.U. ± SD)	Number of Replicates (n)
Tris 10 mM pH 8	663	0.9959	34,095 (±1189)	5
HEPES 10 mM, pH 7.4	620	0.9911	31,313 (±2089)	3
HBSS, pH 7.4	33	0.9841	1695 (±333)	2
PBS pH 7.4	511	0.9994	25,390 (± 2868)	40
PBS pH 8.5	523	0.9988	26,416 (± 1243)	3
PBS pH 9.5	560	0.9985	28,172 (± 2484)	3
PBS–Triton X–100 1%, pH 7.4	780	0.9965	40,045 (±2205)	6
PBS–Triton X–100 1%, pH 8.5	750	0.9904	37,278 (±4863)	6
PBS–Tween 20 1%	7.5	0.9994	379 (±43)	3
PBS–Na–Taurocholate 0.2 mM	534	0.9999	26,662 (±1274)	3
PBS–DMSO 30%	757	0.9947	38,586 (±3190)	3
PBS–0.4 g × L^−1^ BSA, pH 7.4	785	0.9990	39,950 (±1702)	40
PBS–0.4 g × L^−1^ HSA, pH 7.4	718	0.9973	35,823 (±1862)	3

**Table 6 ijms-24-16289-t006:** Associations between categorical parameters and bilirubin levels in plasma of 226 subjects. See population details in Material and Methods.

Categorical Parameter		Median BR (µM) (25–75%)	*p*-Value
**Sex**	Male	5.88 (4.44–7.86)	<0.001
	Female	4.92 (3.31–6.36)	
** *UGT1A1* ** **rs8175347**	––	4.57 (3.77–5.98)	Ref.
	TA–	5.59 (4.02–7.36)	0.008
	TATA	9.38 (5.87–13.71)	<0.001
	TA– + TATA	6.02 (4.31–8.07)	<0.001

## Data Availability

Data are contained within the article.

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
