# Peer review of "Nanoscale Bilirubin Analysis in Translational Research and Precision Medicine by the Recombinant Protein HUG"

_ijms, 2023, doi:10.3390/ijms242216289_

Round 1
Reviewer 1 Report
Comments and Suggestions for Authors
The objective of the study presented by Paola Sist et al. is to demonstrate the suitability of a fluorimetric method for the determination of bilirubin using a recombinant fusion protein linked to the bilirubin-sensor UnaG.
The topic is of relevance for the readers of “International Journal of Molecular Sciences”. Besides the need for an accurate determination of BR levels in biological fluids of humans and veterinary patients, assays that allow the formation of BR in “real time” within a model system, such as cell culture would be highly appreciated by the scientific community. Therefore, the technical idea presented by the authors is very interesting. However, the presented manuscript suffers from some critical deficits.
1) The use of the biosensor UnaG has already been described a useful tool for measuring bilirubin (BR) as is described elsewhere (Sci Rep. 2016 Jun 21:6:28489.doi: 10.1038/srep28489). The fusion to human elastin-like polypeptide (HELP) decreases the sensitivity of UnaG to BR. What is the rational behind using HELP as a fusion component for UnaG? Authors should indicate the advantages of using their fusion protein in comparison to UnaG alone.
2) Authors state that the fusion protein allows determination of very low amounts of BR in plasma/serum, such as mice or rats, or cell culture supernatants. However, they do not bring any evidences for the suitability of their assay regarding these samples.
3) Authors limit the determination of BR to samples with rather diluted levels of BSA (0.4g/L). However, they exquisitely show that HUG-fluorescence strongly depends on the BSA concentration (Fig.3, Fig.4). If the assay should be suitable for determination of BR in plasma or serum of rodents, the protein concentrations tolerated by the assay must be much higher. Best if the assay displays a certain robustness towards the protein concentration, as protein levels may vary in the study objects and animals models. Authors must thoroughly evaluate the sensitivity of the BR determination towards the protein concentration within the sample, particularly for samples displaying low levels of BR.
4) The suitability of the fusion protein for determination of BR in cell culture supernatants must be shown and compared to the method of choice, which is until now is the UV-Vis determination after extraction of BR into a solvent, such as C6H6 or benzene (as decreibed elsewhere: doi: 10.3390/biom5020679 and doi:10.1006/abio.1998.2806.).
5) Authors state that assays allowing the determination of subtle BR amounts would not be available, and that especially determination of heme oxygenase activity would not be expoited by the scientific community (lines 39-43). This statement is absolutely incorrect and documents a striking ignorance of the existing literature:
a. doi: 10.1385/1-59259-054-3:369;
b. doi: 10.1152/ajprenal.00210.2015;
c. doi:10.1016/0003-2697(92)90271-8;
d. doi: 10.1016/j.ab.2004.06.024;
e. doi: 10.1515/hsz-2015-0176;
f. doi: 10.3390/biom5020679;
g. doi: 10.1016/0003-2697(88)90006-1;
h. PMID: 6897023;
i. doi: 10.1016/0003-2697(88)90662-8;
j. doi: 10.1002/0471140856.tx0903s00;
k. doi: 10.1124/dmd.108.025023;
l. doi: 10.1006/abio.1998.2806;
m. doi: 10.1016/j.freeradbiomed.2008.10.044.
Most of these studies used the determination of BR as read out. However, some also exploited quantification of CO, which is produced in equimolar amounts by the HO reaction. Authors are encouraged to incorporate those methods, which have used the determination of BR as a measure for the HO activity (capacity to degrade heme).
6) More information is needed regarding the validation of the quality of the newly described method: for the determination of accuracy it is generally recommended to use at least 5 different concentrations, using different matrices and multiple replicates to calculate the in-run precision, and at least three runs for the determination of the intercurrent accuracy. Additionally, variation coefficients for LOD and LOQ should be added (DOI: 10.17063/bjfs7(4)y2018265). This information could be added to the table A4.
7) Please carefully check the inserted references. Some appear to be inappropriate (#39 (not a suitable reference) and #43 (erroneous citation)).
8) Authors should add to the Materials and Methods section an indication how the human plasma samples were processed for measuring BR concentration (were samples diluted with PBS? How much plasma was used per analysis? Have authors determined the plasma protein concentrations prior to BR determination?).
Comments on the Quality of English LanguageEnglish appears to be OK.
Author Response
Manuscript ijms-2651251
Replies to review report 1
We thank the reviewer for carefully examining our manuscript and raising a number of objections. Please find below our point-to-point replies. Please find the tracked amendments addressing your points in the revised manuscript (with tracked changes of the text).
1) The use of the biosensor UnaG has already been described a useful tool for measuring bilirubin (BR) as is described elsewhere (Sci Rep. 2016 Jun 21:6:28489.doi: 10.1038/srep28489). The fusion to human elastin-like polypeptide (HELP) decreases the sensitivity of UnaG to BR. What is the rational behind using HELP as a fusion component for UnaG? Authors should indicate the advantages of using their fusion protein in comparison to UnaG alone.
Answer
We agree with the reviewer’s remark about the use of UnaG for the direct analysis of bilirubin in human serum. Indeed, we have stated in the Introduction that: “A promising method based on the bilirubin-specific fluorophore UnaG has been developed [22], enabling the direct determination of unconjugated bilirubin concentration in human newborn serum” (page 2, lines 66-68).
As explained in the next paragraph (lines 67-74), we explored the performance of UnaG when fused to a protein scaffold, such as HELP. It is not granted that UnaG fused with other proteins maintains its function. As stated in the Introduction, page 2, lines 74-76): “… it represents a case of an expanded potentiality of UnaG for the direct analysis of bilirubin at the nanoscale level”. We mean that HUG provides an expanded application domain of UnaG is when it is linked on something that may form a solid substrate, such as the extra-cellular matrix protein prototype HELP.
Concerning the lower affinity of HUG for bilirubin than that of UnaG, we have discussed that this is not a relevant limitation in the Discussion (section Advantages and limitations; page 14, lines 437-440): “With respect to UnaG (Kd = 10-10 M), HUG has a lower affinity for bilirubin (Kd = 10-9 M), but this is not a limitation, because it is still higher than BSA or HSA [24], resulting in full extraction of unconjugated bilirubin from the albumin pool”.
Based on these arguments, we think that the current text fully addresses the objections of the reviewer and any other reader.
2) Authors state that the fusion protein allows determination of very low amounts of BR in plasma/serum, such as mice or rats, or cell culture supernatants. However, they do not bring any evidences for the suitability of their assay regarding these samples.
Answer
We have not stated that our study is on measurement of bilirubin in murine serum/plasma or cell cultures. We have stated that:
- the diazo-based method for the determination of bilirubin is inadequate in rodent models/small animals (Abstract, line 13-15).
- the HUG can be applied in buffers at different pH values and in the presence of some additives, as stated in the Results, page 12, lines 356-359: “in various buffered solutions besides PBS, such as Tris pH 8.0 and Hepes pH 7.4. By contrast the Hanks’ balanced salt solution resulted in fluorescence quenching. Supplementation of PBS with detergents, such as Triton X-100 and sodium taurocholate, but not Tween 20, was tolerated”.
- this method was applied in cell cultures and in the isolated perfused rat liver (not plasma) and, adding specific references (ref. 42 and 43, respectively) (Discussion, page 15, lines 459-461).
Specific applications of this assay in human physiological fluids, cells and and tissues, animal sera and so on, cannot be reported in this article that is already very long. Analysis of bilirubin in rat serum/plasma, whose results were obtained by a master student (Donini, L. Microanalysis of bile pigments in the animal serum - Master thesis, University of Trieste, 2021) is currently the topic of a manuscript in preparation. Analysis of bilirubin in cell cultures, whose results were reported in a Conference paper cited in the text (line 451, ref 39) is the topic of a research article submitted to another Journal and currently under revision according to the reviewer suggestions.
Based on these arguments, we think that we cannot add further data to this article.
3) Authors limit the determination of BR to samples with rather diluted levels of BSA (0.4g/L). However, they exquisitely show that HUG-fluorescence strongly depends on the BSA concentration (Fig.3, Fig.4). If the assay should be suitable for determination of BR in plasma or serum of rodents, the protein concentrations tolerated by the assay must be much higher. Best if the assay displays a certain robustness towards the protein concentration, as protein levels may vary in the study objects and animals models. Authors must thoroughly evaluate the sensitivity of the BR determination towards the protein concentration within the sample, particularly for samples displaying low levels of BR.
Answer
We point out that:
- Figure 3 (Correlation of bilirubin analysis by direct UV-Vis spectroscopy versus the diazo colorimetric method). and Figure 4 (Effect of BSA concentration on the UV-Vis spectra of bilirubin) do not show data of HUG fluorescence.
- The range of BSA concentrations tolerated by the HUG assay is 0.4-40 g/l, as shown in Table 3 (Fluorescence intensity of HUG-BR solution as a function of BSA concentration.
Experimental conditions: T = 25 °C and t = 2 h) and commented in the text (Results, pag 7, line 216): “Data show that this assay is independent of BSA concentration (Table 3)”.
Furthermore, immediately after (lines 217-219), we state that: “This result is very important because the same calibration curve, obtained with calibrator solutions with 0.4 g·L-1 BSA can be used to determine the concentration of bilirubin in samples with variable and unknown BSA concentration.”.
Based on these arguments, we think that the objections of the reviewer are addressed in the current text.
4) The suitability of the fusion protein for determination of BR in cell culture supernatants must be shown and compared to the method of choice, which is until now is the UV-Vis determination after extraction of BR into a solvent, such as C6H6 or benzene (as decreibed elsewhere: doi: 10.3390/biom5020679 and doi:10.1006/abio.1998.2806.).
Answer
The HUG assay (as any other assay based on UnaG fluorescence) has the advantage of passing over the extraction of bilirubin from complex biological matrices, thus reducing sources of error as well as the use of health hazardous solvents. As stated in our answer to point 2 above, the method for bilirubin analysis in cell culture media and extracts is published in a conference paper (ref. 42) and detailed in a research manuscript, currently submitted to another journal (revisions ongoing).
Based on these arguments, we think that we cannot add further data to this article.
5) Authors state that assays allowing the determination of subtle BR amounts would not be available, and that especially determination of heme oxygenase activity would not be expoited by the scientific community (lines 39-43). This statement is absolutely incorrect and documents a striking ignorance of the existing literature: [list of references omitted here].
Answer
The reviewer is referring to a list of research articles reporting methods and data of heme oxygenase activity determination. In general, enzyme activity is determined in the presence of an excess, saturating substrate concentration, leading to product formation and analysis under conditions of enzyme Vmax, which is proportional to enzyme concentration. Under physiological conditions, enzymes are not saturated by the substrate and therefore the formation of the product is far below the Vmax. Fine-tuning of homeostasis is also obtained by regulators, which induce relatively small shifts in product formation. Measuring the output of heme oxygenase under homeostatic conditions is completely different than measuring its protein expression by kinetics (Vmax). Measuring heme oxygenase activity modulation under physiological or eve pathological conditions requires highly sensitive detection methods. To avoid any possible misunderstanding of our meaning, we have amended the text as follows (from line 39 on):
“However, direct and sensitive analysis of bilirubin in cells, tissues, blood, and other fluids would allow more precise phenotypic characterization of the effects of HO-1 modulation, and this goal would be more easily attained by the availability of simple, sensitive, and “green” analytical methods that do not use hazardous chemicals and advanced equipment.”
We thank the reviewer for the critique, which prompted us to improve a crucial sentence in the Introduction.
6) More information is needed regarding the validation of the quality of the newly described
method: for the determination of accuracy it is generally recommended to use at least 5
different concentrations, using different matrices and multiple replicates to calculate the in-
run precision, and at least three runs for the determination of the intercurrent accuracy.
Additionally, variation coefficients for LOD and LOQ should be added (DOI:
10.17063/bjfs7(4)y2018265). This information could be added to the table A4.
We followed three guidelines:
- Guideline, I.C.H.H. Bioanalytical method validation and study sample analysis M10. ICH Harmon. Guidel. Geneva, Switz. 2022.
- Ederveen, J. A practical approach to biological assay validation. Hoofddorp Prog. 2010, 106.
- Guideline, I.H.T. Note for Guidance on Validation of Analytical Procedures: Text and Methodology (CPMP/ICH/381/95). Eur. Med. Agency Amsterdam, Netherlands 1995.
To address this objection, we have amended the text by adding the above-mentioned references in Results, 2.2.6 Accuracy and precision, page 11.
“The accuracy of the method was evaluated according to guidelines to be followed for regulatory submissions [31-33]. We analyzed the bilirubin concentration of four different standard solutions (5-10-25-50 nM) and each standard solution was distributed in four wells (n = 4). Analysis was repeated three times, using fresh solutions (n = 3). This protocol was implemented on two different days”.
We thank the reviewer for prompting us to improve our text.
Validation of the method on different solvents is presented in Table 5 (Calibration parameters obtained with nanoscale standard solutions of bilirubin in various solvents). Adaptations to specific matrices is going to be reported case-by-case, as stated above (replies to objections n. 2 and n. 4).
7) Please carefully check the inserted references. Some appear to be inappropriate (#39
(not a suitable reference) and #43 (erroneous citation))
Answer
Reference 39 is a conference abstract published in Investigative Ophthalmology & Visual Science (ISSN:0146-0404E-ISSN:1552-5783), the official online journal of the Association for Research in Vision and Ophthalmology (ARVO). This Journal is indexed in Scopus. We think that this citation is useful, because readers will be able to identify the hopefully forthcoming full article by Tonelotto et al., where details on bilirubin analysis in cell media and extracts is described (see also replies to objections 2 and 4).
Reference 43 is uncorrect in line 278, page 10.
We agree and thank the reviewer for pinpointing this mistake. In amending this citation, we have taken the opportunity to cite again the Ederveen’s document A Practical Approach to Biological Assay Validation (ref. 31), introduced in Results (see above reply to objection n. 6).
8) Authors should add to the Materials and Methods section an indication how the human
plasma samples were processed for measuring BR concentration (were samples diluted
with PBS? How much plasma was used per analysis? Have authors determined the plasma
protein concentrations prior to BR determination?).
Answer
We agree with the reviewer and acknowledge that a subsection of the Materials and Methods was missing. We are very grateful for this remark. We have added the following paragraph in Materials and Methods:
“4.7 Bilirubin Fluorometric measurements in human plasma samples
Human plasma (10 µL) was added to 1.990 mL HUG solution (0.05 mg/mL, pH 7.4) and then divided into two 1-mL aliquots, one for BR and the other for total BR (BR + BR glucuronide) quantification. For analysis of BR, volumes of 200 µL were added directly to the multiwell plate in four replicates. For analysis of total BR, 2.5 µL of ß-glucuronidase was added to the second 1-mL aliquot (final concentration 0.0875 U/µL) and then dis-tributed to the multiwell plate in four replicates. The microtiter plate was incubated at T = 25°C, and fluorescence intensity was measured after 16 hours. Intrinsic sample fluo-rescence was measured in diluted human plasma (5 µl in 1 mL PBS, pH 7.4), and the value (<200 AFU) was subtracted from HUG-BR fluorescence signal (1000-40000 AFU).”
Quantification of plasma proteins is never done in serum/plasma bilirubin determination, which is expressed as mg/dL or µM. In any case, this assay is independent of albumin concentration in a range of 0 – 40 g/L, as shown in Table 3.

Reviewer 2 Report
Comments and Suggestions for Authors
In this study, authors presented a nanoscale fluorometric analysis method with highly specific binding to the recombinant bifunctional protein HELP-UnaG (HUG).
In results, authors provided proper experimental data for show capabilities to new bilirubin assay tools in scientific and commercial use in medicinal fields. For these purposes, authors were examined solvent conditions, stabilizers with traditional uv/vis methods and compared accuracy in new methods. Furthermore, total bilirubin level of human plasma samples were compared and got nice correlational pattern between two assays.
Major comment
- Please describe details to microplate information. Matrix?, coating or ionized?
- and available any grade of BSA?
- Usually, almost fluorescence materials show fluctuation the intensities in each re-measurement points. Please explain about this materials.
- author should discuss about factors of false positive or false negative in scientific/commercial use.
Comments on the Quality of English LanguageN.A
Author Response
Manuscript ijms-2651251
Replies to review report 2
We thank the reviewer for carefully examining our manuscript and raising a number of objections. Please find below our point-to-point replies. Please find the tracked amendments addressing your points in the revised manuscript (with tracked changes of the text).
Major comment
1- Please describe details to microplate information. Matrix?, coating or ionized?
Material
Sterility
Surface Treatment
We have amended the text by adding a last sentence in section 4.1 Materials:
“Black, 96-well microplates (Nunc®, purchased by Thermofisher, code 237107; polystyrene, sterile, non-treated surface)”.
2- and available any grade of BSA?
We have amended the text by adding the >98% purity specification in Materials.
- Usually, almost fluorescence materials show fluctuation the intensities in each re- measurement points. Please explain about this materials.
We have added the following sentence at the end of paragraph 4.5 Fluorometric measurements of bilirubin standard solutions (page 17, line 540).
“The mean fluorescence reading value of empty plate was about 20-30 Arbitrary Fluorescence Unit (AFU), while the intrinsic sample or BSA signal in PBS was about 100-200 AFU and was always subtracted from its own HUG-BR signal (1000-45000 AFU).”
- author should discuss about factors of false positive or false negative in scientific/commercial use
We have added this sentence in the Discussion:
2A distinct advantage of bilirubin assays exploiting UnaG is that there is no interference by ditauro bilirubin, biliverdin, urobilin, hemoglobin and lipids, as demonstrated by pioneering works with UnaG [34][22]. By using HUG, we have confirmed these findings and expanded the spectrum of non-interfering molecules, such as estradiol 17-beta glucuronide and its aglycone, pravastatin, taurocholate, cyanidin 3-glucoside, malvidin 3-glucoside, peonidin 3-glucoside, and resveratrol [43].2
Any other auto-fluorescence signal was subtracted, as specified in the amended text (4.7 Bilirubin Fluorometric measurements in human plasma samples).
We thank the reviewer for the remarks that helped us improving our article.
